# Genome-Wide Identification and Expression Analysis of the MYB Transcription Factor Family in *Salvia nemorosa*

**DOI:** 10.3390/genes15010110

**Published:** 2024-01-17

**Authors:** Huan Yang, Chen Chen, Limin Han, Xiao Zhang, Ming Yue

**Affiliations:** 1The College of Life Sciences, Northwest University, No. 229 Taibai North Road, Xi’an 710069, China; yanghuanmy@stumail.nwu.edu.cn; 2Xi’an Botanical Garden of Shaanxi Province, Institute of Botany of Shaanxi Province, Shaanxi Engineering Research Centre for Conservation and Utilization of Botanical Resources, No. 17 Cuihua South Road, Xi’an 710061, China; chenchen@xab.ac.cn (C.C.); zhxiaoa@163.com (X.Z.); 3College of Life Sciences and Food Engineering, Shaanxi Normal University, Shenhe Avenue, Xi’an 710100, China; hdd_1981_@163.com

**Keywords:** gene family, gene expression, anthocyanin biosynthesis, ornamental *Salvia*

## Abstract

The MYB transcription factor gene family is among the most extensive superfamilies of transcription factors in plants and is involved in various essential functions, such as plant growth, defense, and pigment formation. *Salvia nemorosa* is a perennial herb belonging to the Lamiaceae family, and *S. nemorosa* has various colors and high ornamental value. However, there is little known about its genome-wide *MYB* gene family and response to flower color formation. In this study, 142 *SnMYB* genes (*MYB* genes of *S. nemorosa*) were totally identified, and phylogenetic relationships, conserved motifs, gene structures, and expression profiles during flower development stages were analyzed. A phylogenetic analysis indicated that MYB proteins in *S. nemorosa* could be categorized into 24 subgroups, as supported by the conserved motif compositions and gene structures. Furthermore, according to their similarity with *AtMYB* genes associated with the control of anthocyanin production, ten *SnMYB* genes related to anthocyanin biosynthesis were speculated and chosen for further qRT-PCR analyses. The results indicated that five *SnMYB* genes (*SnMYB75*, *SnMYB90*, *SnMYB6*, *SnMYB82*, and *SnMYB12*) were expressed significantly differently in flower development stages. In conclusion, our study establishes the groundwork for understanding the anthocyanin biosynthesis of the *SnMYB* gene family and has the potential to enhance the breeding of *S. nemorosa*.

## 1. Introduction

*Salvia* represents the largest genus within the Lamiaceae family, encompassing nearly 1000 distinct species of shrubs, herbaceous perennials, and annual plants. This genus has a wide distribution spanning Europe, Asia, Africa, and the Americas [1]. Numerous varieties of Salvia have been employed as traditional medicinal herbs for centuries in the treatment of various ailments. For example, *S. miltiorrhiza* (red sage), which is native to China, has been used to treat and prevent cardiovascular diseases, including coronary heart disease, myocardial infarction (MI), angina pectoris, and atherosclerosis [2]. Moreover, some members of this genus, such as clary sage (*S. sclarea*), Spanish sage (*S. verbenaca*), and Greek sage (*S. triloba* L.), produce essential oils that are rich in terpenes, including linalool acetate, linalool, caryophyllene, and terpineol, which have antimicrobial, anti-amnesic, antidepressant, and anticancer activities as well as other potential health benefits [3,4,5,6,7,8]. Moreover, some species of *Salvia* are grown as ornamental plants for their attractive, colorful flowers and foliage, such as common sage (*S. officinalis*) and scarlet sage (*S. splendens*) [9,10].

*S. nemorosa*, commonly known as “Woodland Sage”, is a hardy, herbaceous perennial plant that is indigenous to broad regions of Central Europe and Western Asia [11]. *S. nemorosa* has dark-purple stems and purple–violet stems borne in long, upright spikes, and its colorful flowers vary from the occasional white to pale pinks, mauves, blues, and purples. Moreover, it blooms from June to October, providing a bounty of sustenance and attracting beneficial insects all summer long. *S. nemorosa* is an easy survival plant to grow in average, moist, and well-drained soil under conditions ranging from full sunlight to partial shade [12]. It prefers sandy or gravelly soil and even tolerates dry soils. It also works well with other perennial plants in a mixed scheme on the fringes of a forest garden or woodland space. Meanwhile, it can attract pollinators and beneficial insects and ensure good pollination rates for other common garden crops. In summary, as a garden plant, *S. nemorosa* is tough and versatile, combining well with many other perennials and grasses and requiring little maintenance.

Anthocyanins are hydrophilic pigments present in plants of the flavonoid group, responsible for the red, purple, and blue hues observed in various fruits, vegetables, flowers, and foliage. Anthocyanins have various functions in plants, including attracting pollinators, protecting against UV radiation, and acting as antioxidants. The biosynthesis of anthocyanins is well understood, and the basic pathway involves the conversion of phenylalanine to 4-coumaroyl-CoA and then, through several enzymatic reactions, compounding chalcone synthase (CHS), chalcone isomerase (CHI), flavanone 3-hydroxylase (F3H), dihydroflavonol 4-reductase (DFR), and anthocyanidin synthase (ANS) to form unstable anthocyanidins, cyanidin, pelargonidin, and delphinidin. Ultimately, a series of stable sugar derivatives are produced by flavonoid 3-O-glucosyltransferase (UFGT), which is responsible for adding sugar molecules to anthocyanidins to form anthocyanins. In addition to the basic genes participating in the biosynthesis pathway, many transcription factors also adjust the accumulation of anthocyanins, the most famous of which is the MYB-bHLH-WD40 complex (MBW complex). Substantial evidence which is obtained from model plants and fruits has demonstrated that the MBW complex, especially the bHLH transcription factor, is a potent regulator of anthocyanin production in higher plants [13,14]. In particular, *MYBs* play an essential part in determining anthocyanin accumulation [15]. For instance, the R2R3-MYB transcription factor *PAP1* (Production of Anthocyanin Pigment 1) in *Arabidopsis thaliana* has been revealed to be able to initiate the expression of *CHS*, *CHI*, *F3H*, *DFR*, and *ANS*, contributing to increased anthocyanin accumulation [16]. Similarly, the over-expression of *AtMYB124* resulted in an increase in anthocyanin accumulation in Arabidopsis leaves and flowers, while knocking it out reduced anthocyanin levels. Additionally, *AtMYB124* could interact with *PAP1* and activate it to improve the expression of anthocyanin biosynthesis genes [17]. In strawberry, *FaMYB5*, a newly identified R2R3-MYB transcription factor, positively regulates the biosynthesis of anthocyanin and proanthocyanidin [18]. Notably, *LvMYB5* can combine and activate the promoter of the *ANS* gene to increase anthocyanin levels, while *LvMYB1* inhibits anthocyanin synthesis in lily flowers [19]. Overall, *MYBs* in plants are multifunctional agents that influence anthocyanins as positive or negative regulators through fine-tuning the expression of target genes.

The MYB gene family, one of the largest families of transcription factors in higher plants, is distinguished by a preserved MYB DNA-binding domain located at the N-terminus [20]. Effectively, it comprises three conserved functional domains: a DNA-binding domain (DBD), a transcriptional activation domain (TAD), and an incompletely defined negative regulatory region (NRD) [21]. According to the number of conserved structural domains, the MYB family can be categorized into four distinct subfamilies, namely, MYB-related (1R-MYB), MYB-R2R3 (2R-MYB), MYB-R1R2R3 (3R-MYB), and MYB-4R [22]. To date, many MYB families have been identified at the genome-wide level, including *Arabidopsis thaliana* [23], watermelon [24], *Solanum lycopersicum* [25], rice [26], maize [27], soybean [28], and poplar [29], and the regulation functions of *MYBs* have been further studied. However, there is limited knowledge regarding the composition of the MYB gene family in *S. nemorosa* and its involvement in the regulation of flower pigmentation patterns. Based on the above points, we first identified a total of 142 *SnMYB* genes (*MYB* genes of *S. nemorosa*) and characterized 10 candidate genes according to their similarity with *AtMYB* genes related to the regulation of anthocyanin biosynthesis. Then, the gene structures, chromosome distribution, conserved motifs, cis-elements, and expression profiles of candidate *MYB* genes were systematically analyzed. This research not only offers a point of reference for further elucidating the roles of MYB genes but also contributes to a deeper understanding of the diverse functions of MYBs in *S. nemorosa*.

## 2. Materials and Methods

### 2.1. Genome-Wide Identification of MYB Transcription Factor Genes of S. nemorosa

The *S. nemorosa* genome sequences were assembled and annotated, and the database was saved in our lab (unpublished data). We downloaded all known *MYB* genes of *Arabidopsis thaliana* from TAIR library (http://www.Arabidopsis.org/, accessed on 10 October 2023) [30]. These sequences were utilized as queries when employing the BLASTP online tool (with an E-value threshold of ≤1 × 10^−5^) [31] to identify *S. nemorosa* MYB family members within our *S. nemorosa* genome database. Meanwhile, the Hidden Markov Model (HMM) profile for *MYB* was downloaded from Pfam database (http://pfam.sanger.ac.uk/, accessed on 12 October 2023), and an HMM search was performed using a local version of the HMMER program with the default values [32]. Then, the *MYB* genes were manually identified in Pfam database, and the sequences without complete MYB domains were discarded. Then, the candidate protein sequence containing MYB domains (PF000249) were confirmed again based on the Conserved Domain Database (CDD) of NCBI (https://www.ncbi.nlm.nih.gov/cdd/, accessed on 14 October 2023) [33]. The finalized 142 *SnMYB* genes were identified and renamed SnMYBs. Moreover, the features of SnMYB proteins were predicted using the ExPaSy Proteomics Server (http://www.expasy.org/sprot/sp-docu.html, accessed on 18 October 2023).

### 2.2. Phylogenetic Analysis and Functional Prediction of SnMYB Genes

The full protein sequences of *SnMYBs* and *AtMYBs* were aligned using the MUSCLE product produced by MEGA 11 (https://www.megasoftware.net/, accessed on 20 October 2023), and conservation regions were analyzed using Esprint (https://espript.ibcp.fr/ESPript/cgi-bin/ESPript.cgi, accessed on 22 October 2023). A phylogeny tree was constructed via the neighbor-joining (NJ) method with 1000 bootstraps. Then, using the iTOL web browser (https://itol.embl.de/, accessed on 23 October 2023), the visualization of the phylogenetic tree was performed.

### 2.3. Gene Structure, Chromosomal Distribution, and Conserved Domain Analysis of SnMYB Genes

The DNA and cDNA sequences corresponding to each predicted *MYB* gene were obtained from our genome database. Maps of the intron/exon structures and chromosome locations of *MYB* genes were made using the TBtools 1.06876 software product (https://github.com/CJ-Chen/TBtools/releases, accessed on 24 October 2023) [34]. The conserved motifs were predicted using the MEMEs program (https://meme-suite.org/meme/tools/meme, accessed on 25 October 2023). Protein tertiary structure predictions of SnMYBs were performed on the Alphafold website (https://alphafold.ebi.ac.uk/, accessed on 27 October 2023), and domain analyses (http://smart.embl-heidelberg.de/, accessed on 29 October 2023) were performed on all the obtained protein sequences. The distribution of *MYB* genes in the chromosomes of *S. nemorosa* was determined using the gff annotation file and gene density file of the genome in the Gene Location Visualize tool in the GTF/GFF function module of TBtools (https://github.com/CJ-Chen/TBtools/releases, accessed on 30 October 2023), and TBtools was also used to analyze the physicochemical properties of proteins.

### 2.4. Subcellular Localization and cis-Elements of SnMYB Genes

The protein sequences of SnMYBs were used to predict subcellular localization on the WoLF PSORT website (https://wolfpsort.hgc.jp/, accessed on 1 November 2023). Then, we obtained the promoter sequences about 2000 bp upstream from the start codon of each candidate *SnMYB* gene, and the prediction of the cis-acting elements was performed using PlantCARE website (http://bioinformatics.psb.ugent.be/webtools/plantcare/html/, accessed on 4 November 2023) within these regions.

### 2.5. Transcriptome Sequencing

To determine the expression patterns of SnMYBs during the four flower development stages (denoted as P1–P4, corresponding to bud stage, initial flowering stage, full-flowering stage, and end-flowering stage, with different degrees of purple coloration) were collected for RNA extraction using the RNAiso kit (TaKaRa, Tokyo, Japan). RNA extraction was conducted based on the manufacturer’s protocol. The Illumina RNA-Seq library was constructed using the Illumina HiSeq 2500 platform produced by Biomark Bioinformatics Co., Ltd. (Beijing, China). The expression profiles of SnMYB genes at P1-P4 flower stages were assessed based on their FPKM (fragments per kilobase of exon model per million mapped fragments) values, and a heat map was constructed using TBtools 1.06876.

### 2.6. Quantitative Real-Time PCR

The expression patterns of 10 selected *SnMYBs* genes were further examined using qRT-PCR. The total RNA extracted from samples P1–P4 using the RNAiso kit and was used to carry out reverse transcription using the PrimeScript RT reagent Kit (Takara). Primers were designed using PrimerPermier 3 plus software (http://www.primer3plus.com/cgi-bin/dev/primer3plus.cgi, accessed on 7 November 2023), and *Actin* was used as reference gene (Appendix A). The FastStart Universal SYBR Green kit (Roche, located in Shanghai, China) was used along with a 20 μL reaction system for qRT-PCR analysis via an LightCycler 96 instrument (Roche, located in Shanghai, China). Three biological and technical replicates were used. The 2^−ΔΔct^ method was utilized to calculate the relative expression levels of *SnMYB* genes.

## 3. Results

### 3.1. Identification of Members of SnMYB Family

To identify *MYB* transcription factor genes in *S. nemorosa* (*SnMYBs*), we performed BLAST and HMM searches. A total of 142 *SnMYB* genes were identified. Furthermore, we calculated the basic physicochemical parameters of SnMYB proteins. The length of the SnMYB protein ranged from 189 to 991 aa, while the molecular weight ranged from 21,454.17 Da to 110,231.84 Da. Their isoelectric points were between 4.46 and 10.21, and the instability index ranged from 34.29 to 81.16 (Table 1). Furthermore, an analysis predicting subcellular location indicated that all of them were situated within the nucleus.

### 3.2. Phylogenetic Analysis and Classification of the MYB Gene Family

In order to clarify the evolutionary relationships and possible roles of SnMYBs, a neighbor-joining phylogenetic tree was generated by comparing sequences of SnMYBs and AtMYBs (Figure 1). The findings revealed that the SnMYBs could be categorized into 24 distinct clades. (Figure 1). The clustered genes of one clade usually displayed conserved functions in plants [35]. We found that the reported *AtMYBs* associated with anthocyanin synthesis, such as *AtMYB113*, *AtMYB114*, *AtMYB90*, *AtMYB75*, *AtMYB123*, *AtMYB12*, *AtMYB11*, and *AtMYB111*, were clustered with SnMYBs in functional categories [30,36,37]. For example, in clade X, *Sne05G053740.1 (SnMYB90)*, *Sne05G053730.1 (SnMYB114)*, *Sne05G053750.1 (SnMYB113b)*, *Sne05G053810.1 (SnMYB75)*, *Sne05G053870.1 (SnMYB1)*, and *Sne05G053820.1 (SnMYB113a)* were clustered together with *AtMYB113*, *AtMYB114*, *AtMYB90*, and *AtMYB75*, which were reported to be the transcriptional regulators of anthocyanin biosynthesis. On the one hand, the overexpression of *AtMYB113* or *AtMYB114* led to increased pigment production; on the other hand, *AtMYB* gene expression was down-regulated and anthocyanin was obviously deficient in plants due to harboring RNAi constructs targeting *AtMYB113* or *AtMYB114*, indicating that *MYB113* or *MYB114* could regulate anthocyanin synthesis [16]. Alternatively, when *AtMYB75* and *AtMYB90* were co-transfected with any of four R/B-like bHLH proteins, they were found to activate transcription via the *DFR* promoter in reporter genes. It was observed that the two proteins significantly enhanced the expression of the *AtDFR* promoter, thereby suggesting their potential role in regulating the expression of genes involved in anthocyanin biosynthesis [38]. Similarly, according to our results, *Sne01G015730.1 (SnMYB82)*, *Sne05G076150.1 (SnMYB113a)*, *Sne03G025760.1 (SnMYB111)*, and *Sne05G066770.1 (SnMYB6)* were speculated to be associated with anthocyanin regulation. It has been reported that *Arabidopsis MYB123 (TT2)* is involved in the biosynthesis of proanthocyanin (PA) via combining with TT2 (MYB) and TT8 (bHLH), forming an MYB/bHLH complex, which plays a part in TTG1-dependent regulatory pathways [39]. AtMYB123 is responsible for inducing the expression of the BAN gene, which encodes the essential enzyme for proanthocyanidin biosynthesis in the outer seed coat of *A. thaliana*, which plays a regulatory role in the metabolism of anthocyanins. To focus on the study of *SnMYBs* related to anthocyanin regulation, we conducted further analyses on the above 10 candidate *SnMYB* genes.

### 3.3. Conserved Sequence Analysis and Conserved Motifs of Candidate MYB Proteins

In order to investigate the preservation of the sequence, an analysis was conducted. A prior investigation revealed that the distinctive feature of the MYB domain was a series of evenly spaced and strongly conserved tryptophan residues (W), consistent with the patterns of R2 [-W-(X19)-W-(X18/19)-W-] and R3 [-F/L-(X18)-W-(X18)-W-] repeats [40]. As we observed, the sequence alignment illustrated that the conserved residues Trp^17^, Trp^37^, and Trp^57^ in the R2 domain and Trp^89^ and Trp^108^ in the R3 domain, as well as the peptide chain between them, to form a hydrophobic core (Figure 2). The MYB proteins were characterized by a highly conserved DNA-binding domain, comprising one to four helix–turn–helix (HTH) structures and functioning as tandem repeats referred to as R0R1R2R3 within an MYB protein [41]. Meanwhile, the R2 and R3 regions were made up of well-defined α-helices and β-turns further constituting a helix–turn–helix (HTH) structure [42]. To better understand the features of the DNA-binding regions of candidate SnMYB genes, we utilized AlphaFold to forecast the three-dimensional shapes of the proteins produced by 10 candidate genes (Figure 3). These structures were in accordance with the reported characteristics of typical plant R2R3-MYB proteins [43].

Furthermore, a set of 10 motifs was chosen from the potential *S. nemorosa* R2R3-MYB sequences. Analysis of the motifs indicated that the majority of the R2R3-MYBs harbored motifs 1, 2, 4, 5, and 6, underscoring the highly conserved nature of the R2R3-MYB structures, as depicted in Figure 4A. Among them, the motif compositions of SnMYB82 and SnMYB12 were quite different, and this may be related to their special functions.

### 3.4. Analysis of Gene Structure, Chromosome Distribution, and cis-Element of Candidate MYB Genes

The gene structure results indicated that the 10 candidate *SnMYBs* contained two introns and more than four CDS sequences (Figure 4B). Among them, only *SnMYB114*, *SnMYB90*, *SnMYB6*, and *SnMYB111* contained URT sequences, and *SnMYB113a*, *SnMYB113b*, *SnMYB1*, *SnMYB75*, *SnMYB82*, and *SnMYB12* had relatively similar gene structures: they all had six CDS sequences separated by two introns.

In addition, the distribution of the candidate *MYBs* in chromosomes was detected (Figure 5). Ten *SnMYBs* were distributed unevenly across six *S. nemorosa* chromosomes; among them, Chr5 had the greatest length, more than 100 Mb, while Chr6 was the shortest, not exceeding 60 Mb in *S. nemorosa*. The majority of the SnMYB genes were clustered at both ends of their respective chromosomes, with a smaller number of genes located in the central region of the chromosome, wherein *SnMYB82* was distributed across Chr1, *SnMYB111* was distributed across Chr3, and *SnMYB113a*, *SnMYB6*, *SnMYB90*, *SnMYB114*, *SnMYB113b*, *SnMYB75*, *SnMYB1*, and *SnMYB113a* were distributed across Chr5.

Moreover, in order to identify the regulatory components of the SnMYBs, we examined the 2000-base-pair (bp) upstream sequences of the coding region of the *SnMYB* genes, which were utilized for the prediction of their cis-acting elements. (Figure 6). A total of 31 cis-regulatory elements of *SnMYB* genes were predicted; among them, six were related to cellular development, including root-specific, the differentiation of the palisade mesophyll cells, seed-specific regulation, endosperm expression, and flavonoid biosynthetic genes regulation. Seven hormone-related cis-regulatory elements were also identified, including MeJA-responsiveness, gibberellin-responsive, abscisic acid, and auxin. Similarly, thirteen cis-elements associated with stress were also found, including light-responsive elements, drought inducibility, low-temperature responsiveness, anaerobic induction, defense, and stress responsiveness. Most of the *SnMYB* promoters contained various hormone-responsive elements and stress-responsive elements, including MeJA responsiveness, light-responsive elements, and abscisic acid responsiveness, which were present in all the *SnMYB* promoters, demonstrating that the three elements had indispensable roles, and the genes may take part in responses to multiple hormones and stress [44]. Only *SnMYB82* contained the elements of circadian control, differentiation of the palisade mesophyll cells, and seed-specific regulation, indicating that *MYB82* might influence the growth and development of *S. nemorosa*. In addition, a root-specific element, an endosperm expression element, a flavonoid biosynthetic gene regulation element, and anoxic specific inducibility only existed in *MYB90*, *MYB111*, *MYB1*, and *MYB6*, respectively. These results reveal that the differences in the transcriptional regulation of *SnMYBs* indicate the diversity of *SnMYB* functions.

### 3.5. Expression Pattern of Candidate SnMYBs

Based on the *S. nemorosa* transcriptome data, the expression pattern of the *MYB* gene family was evaluated. The results manifested that five *SnMYBs*, namely, *SnMYB6*, *SnMYB90*, *SnMYB114*, *SnMYB111*, and *SnMYB12*, presented certain expression differences (Figure 7A). Among them, *SnMYB6* had the highest expression in P1, P2, and P3, while *SnMYB90* had the highest expression in P4. However, the expression levels of *SnMYB113a*, *SnMYB113b*, *SnMYB75*, *SnMYB1*, and *SnMYB82* were very low (Figure 7A, Appendix A).

Subsequently, the expression levels of *SnMYB* genes were measured using qRT-PCR (Figure 7B) at four different stages of flower development. As can be seen from Figure 7B, there were no prominent expression transformations of *SnMYB1*, *SnMYB113a*, *SnMYB113b*, and *SnMYB114* at four stages. On the contrary, *SnMYB6*, *SnMYB12*, *SnMYB75*, *SnMYB82*, *SnMYB90*, and *SnMYB111* had higher expression levels. In the different stages of flower development, the expression level of *SnMYB6*, *SnMYB12*, and *SnMYB90* increased and reached a peak at the P4 stage. But as for *SnMYB82* and *SnMYB111*, the expression level decreased throughout flower growth. The gene expression level of *SnMYB75* remained relatively high in all four stages.

## 4. Discussion

In plants, ample studies have been conducted, indicating that the MYB-bHLH-WD40 constitutes an “MBW” complex which regulates the anthocyanin metabolic biosynthesis pathway [45]. A previous study on *Malus pumila Mill.* indicated that the *MdMYB3* gene not only regulates the accumulation of anthocyanin in peels but also participates in the regulation of flower development, particularly pistil development [46]. In *Arabidopsis*, *AtMYB113* and *AtMYB114* positively regulated anthocyanin biosynthesis [47]; on the contrary, *AtMYB4* and *AtMYBL2* take part in the suppression of flavonoid accumulation [48]. In petunia, *AN2*, *AN4*, *DEEP PURPLE (DPL)*, and *PURPLE HAZE (PHZ)* were thought to be the members of the “MBW” complex which could influence anthocyanin accumulation [49]. However, the role of *MYB* genes in *S. nemorosa* how to regulate anthocyanin biosynthetic pathways is currently unknown. Therefore, to provide comprehensive insights into the fundamental characteristics of the MYB gene family in *S. nemorosa*, we conducted an analysis that involved the identification of 142 MYB gene family members from the *S. nemorosa* genome. This analysis encompassed an examination of their gene structures, phylogenetic relationships, chromosomal localization, cis-acting elements, and expression patterns, providing relevant information for future research efforts focused on understanding the processes involved in the creation and control of anthocyanins during the flower development stages in *S. nemorosa*.

Phylogenetic analysis is a reliable method for identifying the functions of genes, as homologous proteins that cluster in the same clade possess similar or identical functions due to their similar structures, and vice versa [50]. In *Arabidopsis*, a mass of *MYB* genes or proteins had been functionally characterized. For example, *AtMYB33* and *AtMYB65* redundantly facilitated anther development, but this developmental decision was greatly influenced by environmental factors [51]. *AtMYB112* can accelerate anthocyanin accumulation during subjection to salinity and high light stress, constituting a regulator that promotes anthocyanin cumulation in abiotic stress situations [52]. *AtMYB115* and *AtMYB118* could regulate the genes in the evolution of a novel BZ-GLS (benzoyloxy glucosinolate) pathway in *A. thaliana*, which might be negative regulators in GLS biosynthesis [53]. Thus, it provided the opportunity to predict the functions of *SnMYB* genes through phylogenetic analysis and sequence comparisons with their *Arabidopsis* homologous sequence. In this study, there were ten genes clustered into the corresponding clade that had been reported to regulate anthocyanin synthesis in *Arabidopsis*, demonstrating that they might work in a similar way to regulate anthocyanin synthesis. Thus, we chose to conduct a comprehensive bioinformatics analysis on the ten candidate genes, with the aim of offering valuable insights for future investigations into the regulation of anthocyanin MYB genes.

To evaluate the expression patterns of the 10 candidate genes during flower development, RNA-seq and qRT-PCR were employed. As a result, the expression level analysis demonstrated that *SnMYB6*, *SnMYB12*, *SnMYB90*, and *SnMYB111* were related to anthocyanin biosynthesis, and *SnMYB6*, *SnMYB12*, and *SnMYB90* had high levels of expression in late flower growth (P4). This result is consistent with the deepest purple coloration of the petals during the P4 stage. Numerous pieces of evidence suggest a correlation between the expression level of the MYB gene and the accumulation of anthocyanin content, indicating its regulatory function in anthocyanin synthesis. It has been reported that *MYB12* could be involved in controlling the expression of early biosynthetic genes (EGBs), while the regulation of the late biosynthetic genes (LGBs) demands the participation of *MYB90*, and *MYB6*, resulting in the improvement of anthocyanin and proanthocyanidin accumulation [54,55]. But the expression level of the *SnMYB111* gene was highest in early flower development (P1) and decreased throughout flower growth, leading to speculation that it may play a negative regulatory role in the accumulation of anthocyanins. Furthermore, in *Arabidopsis*, MYB113 could bind directly to the promoter sequence of *CHS*, *F3H*, and *FLS1*, demonstrating that MYB111 plays a positive role in flavonoid biosynthesis regulation, and the overexpression of MYB111 could improve tolerance to salt stress [56]. These two instances demonstrate that MYB111 displays species-specific characteristics in various plants and may fulfill distinct functions. Overall, our results indicate that *SnMYB6*, *SnMYB12*, and *SnMYB90* play a positive role, in contrast, *SnMYB111* might play a negative role in the control of anthocyanin accumulation in *S. nemorosa* flowers. It is noteworthy that there were contrasting findings regarding the expression patterns of *SnMYB75* and *SnMYB82* as observed. While these genes exhibited high expression levels as determined via qRT-PCR, they were found to be almost undetectable in the RNA-seq data. This disparity may be attributed to potential preferential amplification interference during the sequencing process, which could impact the accuracy of gene expression results. Consequently, alternative methods will be employed to validate the expression levels of these two genes. Moreover, this study is constrained by its exclusive assessment of the expression pattern of the MYB gene through qRT-PCR without experimental confirmation of its gene function. As a result, our future investigations will involve analyzing the selected genes, including SnMYB90, SnMYB6, and SnMYB75, through stable overexpression, gene-editing, and silencing experiments. Additionally, transient expression system, yeast one-hybrid, and EMSA experiments will be utilized to evaluate the transcriptional regulatory function of the MYB transcription factor for downstream anthocyanin biosynthetic enzyme genes [57,58].

Moreover, cis-regulatory elements, acting as momentous molecular switches, were involved in the regulation of gene transcription when encountering external stimuli [59]. To illuminate the functions of *SnMYB* genes, six cellular-development-related cis-regulatory elements, seven hormone-related cis-regulatory elements, and thirteen stress-related cis-elements were identified in *SnMYB* promoter regions by analyzing cis-regulatory elements in the promoter regions. The ten *SnMYB* genes contained diverse types and vast quantities of cis-acting regulatory elements in each promoter region, such as elements involved in the MeJA-response, the salicylic acid response, the gibberellin response, the low-temperature response, and light response, as well as a drought-induced MYB-binding site, which have been reported to be involved in the processes of growth metabolism and environmental stress response [60,61].

## 5. Conclusions

In the current study, we recognized 142 high-confidence *S. nemorosa* MYB genes through a genome-wide survey and executed bioinformatics analyses to divulge their physiochemical properties and phylogenetic relationships. Moreover, the transcriptional expression levels of 10 R2R3-type *SnMYB* genes at four flower development stages were analyzed, and the association with the anthocyanin expression levels of the *SnMYB6*, *SnMYB12*, *SnMYB90*, *SnMYB75*, and *SnMYB82* genes was considered for further functional characterization. In conclusion, this study provides important candidate genes for anthocyanidin biosynthesis research on *S. nemorosa* and makes a contribution to improving the potential application of *S. nemorosa* in breeding.

## Figures and Tables

**Figure 1 genes-15-00110-f001:**
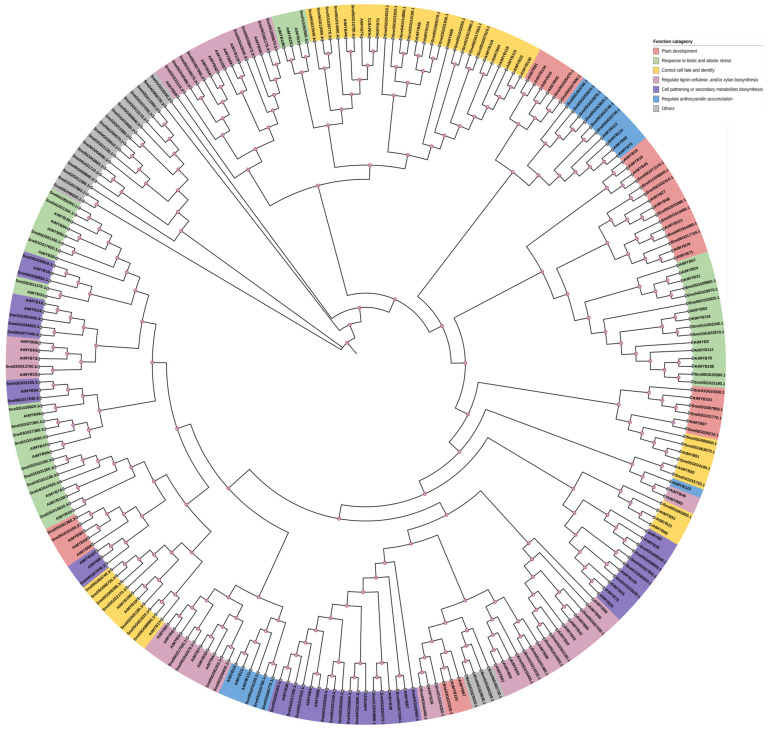
Phylogenetic tree of MYB genes from *S. nemorosa* and *Arabidopsis thaliana*. The maximum-likelihood tree was generated using MEGA11 with 1000 bootstrap replicates. The complete nucleotide sequences of 142 MYB genes in *S. nemorosa* and 119 *Arabidopsis thaliana* MYB genes were used.

**Figure 2 genes-15-00110-f002:**
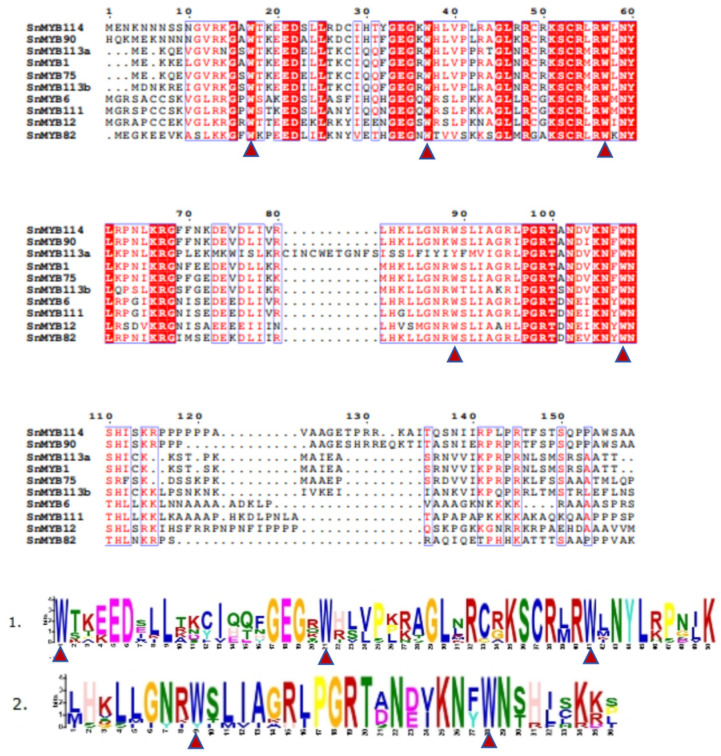
Predicted R2 and R3 domains of candidate *S. nemorosa* MYB proteins and their sequence logos. The bit score serves as a measure of the information content at each position within the sequence, while the red triangle below denotes the conserved tryptophan residues (Trp, W).

**Figure 3 genes-15-00110-f003:**
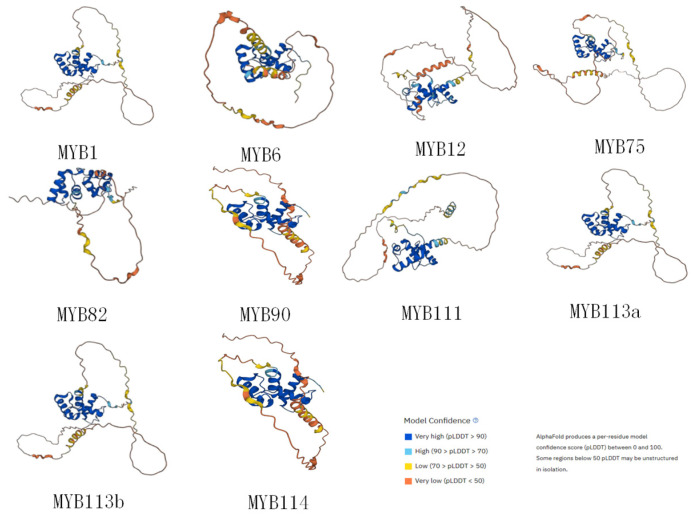
Tertiary structure predictions of the 10 candidate proteins.

**Figure 4 genes-15-00110-f004:**
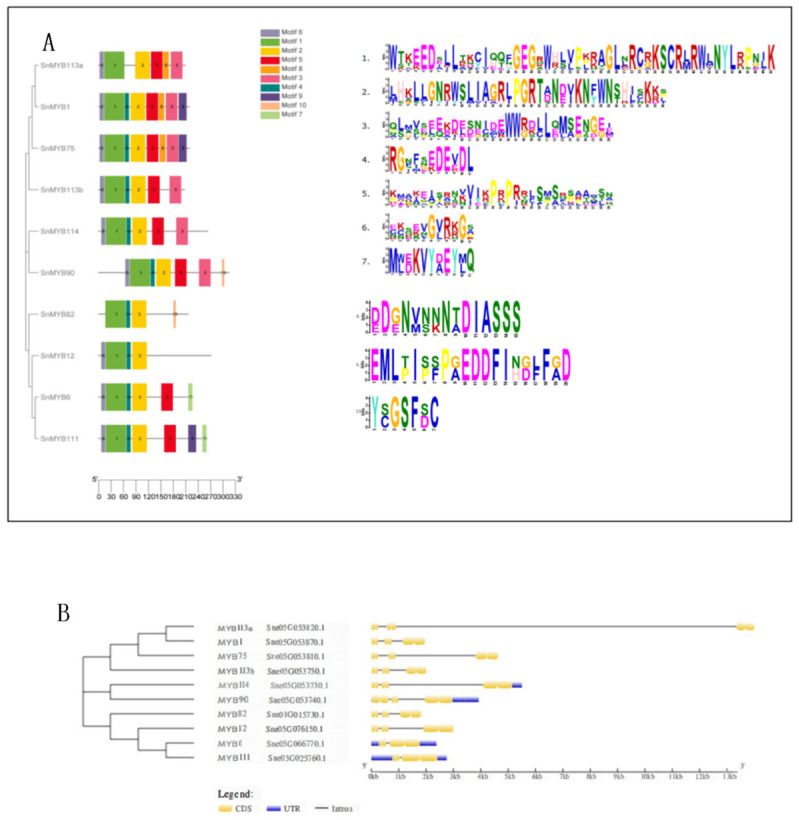
Phylogenetic relationships and gene structures and architectures of the conserved protein motifs and their sequence logos for candidate *S. nemorosa MYB* genes. (**A**) Motif composition of 10 *S. nemorosa* MYB proteins. The motifs numbered 1–10 are displayed in various colored boxes. (**B**) The gene structures of candidate *S. nemorosa MYB* genes, blue boxes indicate UTR, yellow boxes stand for CDS, and black lines indicate introns. The number manifests the phases of the relevant intron.

**Figure 5 genes-15-00110-f005:**
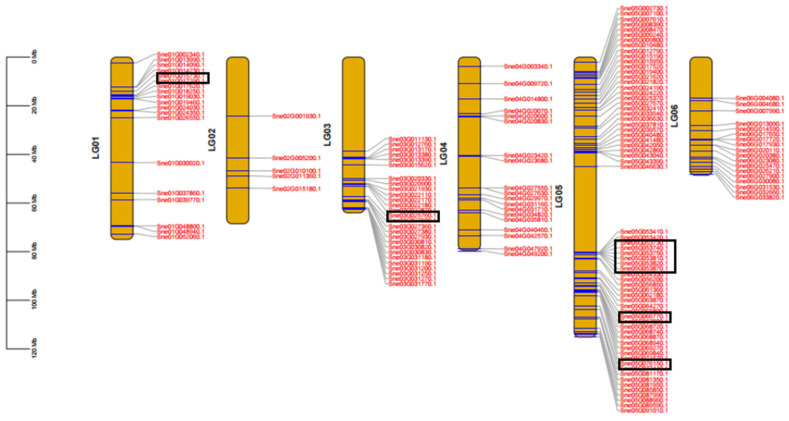
Chromosome localization of the *MYB* gene family in *S. nemorosa*. SnMYB genes were distributed unevenly across six *S. nemorosa* chromosomes, of which the black boxes represented 10 candidate gene sequences.

**Figure 6 genes-15-00110-f006:**
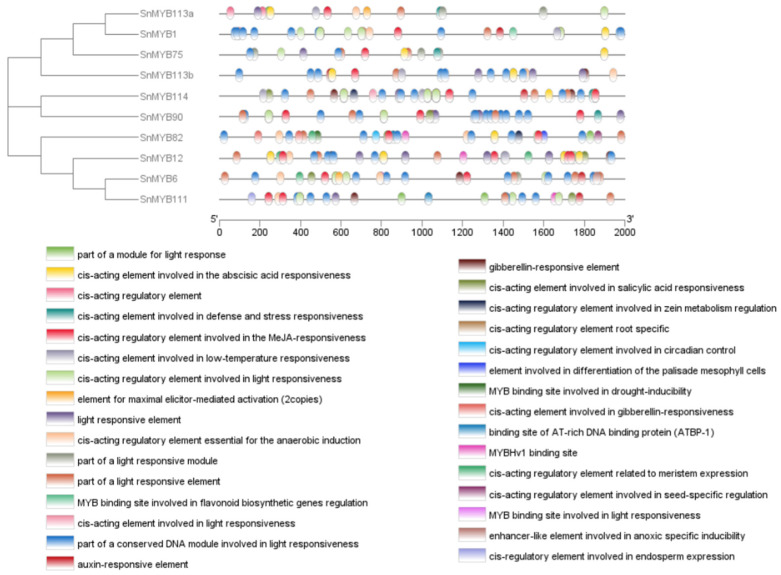
Visualization of promoter cis-acting element prediction of *MYB* genes in *S. nemorosa*. Promoter sequences (−2000 bp) of *SnMYB* genes were analyzed using PlantCARE. Blocks of different colors represent different elements.

**Figure 7 genes-15-00110-f007:**
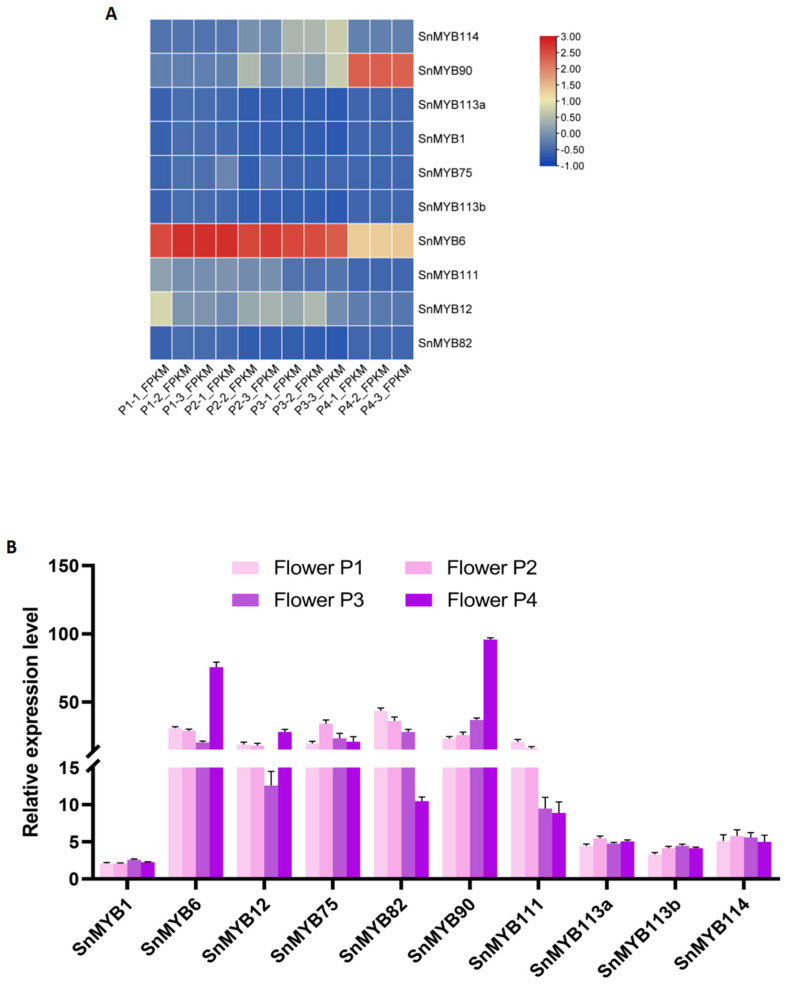
Gene expression patterns of 10 candidate *S. nemorosa MYB* genes. (**A**) Heatmap of expression profiles of candidate *MYB* genes differentially expressed in the petals at four stages of flower development with different degrees of purple coloration (marked as P1–P4) were collected from *S. nemorosa*. (**B**) The expression levels of candidate *MYB* genes in four flower development stages were investigated using qRT-PCR.

**Table 1 genes-15-00110-t001:** Information on 142 MYB gene family members in *Salvia nemorosa*.

Sequence ID	Number of Amino Acids	Molecular Weight	Theoretical pI	Instability Index	Aliphatic Index	Grand Average of Hydropathicity	Subcellular localization
Sne05G042860.1	278	30,211.01	9.51	74.17	65.22	−0.522	Nucleus
Sne03G030830.1	212	24,720.50	5.69	62.63	68.07	−0.808	Nucleus
Sne03G031180.1	292	33,194.90	5.82	51.35	75.27	−0.695	Nucleus
Sne05G061360.1	279	30,929.09	6.30	46.43	79.53	−0.516	Nucleus
Sne03G031250.1	249	28,243.39	5.73	58.81	62.73	−0.769	Nucleus
Sne04G040460.1	266	29,882.20	7.75	68.00	83.68	−0.414	Nucleus
Sne05G063870.1	302	35,080.02	9.33	65.78	69.54	−0.924	Nucleus
Sne01G014230.1	292	33,358.31	5.71	56.71	62.81	−0.712	Nucleus
Sne06G032660.1	289	31,985.97	8.10	54.61	66.99	−0.558	Nucleus
Sne01G019460.1	281	30,878.56	5.81	52.26	68.40	−0.720	Nucleus
Sne03G013380.1	331	36,176.32	6.01	59.31	68.94	−0.761	Nucleus
Sne03G031770.1	403	44,076.92	5.66	55.81	65.98	−0.597	Nucleus
Sne01G013990.1	772	88,384.89	9.00	42.44	60.13	−0.917	Nucleus
Sne04G020600.1	242	26,989.37	6.38	59.53	72.64	−0.482	Nucleus
Sne05G017520.1	253	28,577.14	8.74	58.53	60.20	−0.874	Nucleus
Sne02G010100.1	986	109,507.46	5.00	58.51	62.34	−0.689	Nucleus
Sne05G054330.1	232	25,779.50	9.05	44.63	74.61	−0.450	Nucleus
Sne06G031530.1	298	32,573.52	8.10	56.86	64.97	−0.556	Nucleus
Sne05G032410.1	431	47,921.07	4.89	59.11	84.76	−0.283	Nucleus
Sne01G030020.1	220	24,994.85	7.95	56.18	59.86	−0.823	Nucleus
Sne03G021930.1	432	48,916.86	7.55	49.67	69.10	−0.777	Nucleus
Sne04G020070.1	336	37,817.36	8.32	47.99	55.27	−0.907	Nucleus
Sne04G042570.1	277	30,510.32	8.94	54.23	67.04	−0.604	Nucleus
Sne05G056200.1	305	33,880.80	6.61	44.83	65.97	−0.719	Nucleus
Sne05G068740.1	287	32,253.59	8.99	56.46	74.84	−0.640	Nucleus
Sne05G053820.1	209	24,201.66	9.10	57.97	69.00	−0.693	Nucleus
Sne05G019400.1	379	42,110.86	6.52	51.67	64.17	−0.661	Nucleus
Sne03G022870.1	228	26,638.96	7.01	66.83	67.63	−0.811	Nucleus
Sne03G015620.1	279	31,737.76	8.76	55.96	73.08	−0.661	Nucleus
Sne05G041480.1	304	34,581.67	6.37	45.23	65.46	−0.740	Nucleus
Sne04G031710.1	295	31,585.68	9.61	60.94	72.07	−0.358	Nucleus
Sne05G002730.1	339	37,751.80	6.08	44.08	70.71	−0.669	Nucleus
Sne05G007610.1	362	41,749.46	5.98	64.56	61.16	−0.916	Nucleus
Sne03G011130.1	297	33,032.67	8.70	54.56	50.61	−0.980	Nucleus
Sne03G022170.1	274	31,367.48	9.18	57.96	69.85	−0.805	Nucleus
Sne05G012790.1	316	35,109.66	9.06	53.49	67.06	−0.679	Nucleus
Sne05G069270.1	461	51,451.44	5.87	51.73	65.64	−0.754	Nucleus
Sne05G068870.1	308	34,738.68	5.69	57.15	54.61	−0.726	Nucleus
Sne06G007990.1	224	26,608.70	9.56	67.67	63.57	−1.071	Nucleus
Sne05G042050.1	382	42,406.41	6.11	55.73	66.70	−0.659	Nucleus
Sne05G071570.1	195	22,629.76	8.62	52.34	77.59	−0.646	Nucleus
Sne03G013170.1	329	36,608.42	8.74	45.49	67.33	−0.704	Nucleus
Sne06G017720.1	230	26,767.91	8.97	62.31	70.04	−0.965	Nucleus
Sne04G027550.1	396	43,802.19	4.71	48.64	82.50	−0.519	Nucleus
Sne03G030820.1	248	28,255.59	6.45	81.16	60.60	−0.715	Nucleus
Sne05G091010.1	243	27,155.76	9.03	53.75	67.53	−0.750	Nucleus
Sne05G007100.1	259	30,019.18	6.09	62.80	76.02	−0.628	Nucleus
Sne02G015180.1	271	30,441.58	5.98	49.65	65.68	−0.710	Nucleus
Sne04G023420.1	297	33,832.75	6.46	57.41	64.07	−0.757	Nucleus
Sne04G035810.1	336	37,778.40	8.91	51.12	54.40	−0.895	Nucleus
Sne01G014090.1	206	23,702.65	8.85	48.89	63.40	−0.744	Nucleus
Sne01G015730.1	216	24,632.99	7.78	52.67	75.46	−0.665	Nucleus
Sne03G020900.1	212	24,644.11	9.06	42.03	67.17	−0.754	Nucleus
Sne01G024030.1	199	22,749.71	9.28	50.00	59.90	−0.841	Nucleus
Sne05G015950.1	259	29,167.76	5.36	46.41	76.41	−0.585	Nucleus
Sne03G027360.1	200	23,032.02	9.61	52.60	68.25	−0.957	Nucleus
Sne06G033820.1	269	31,164.72	5.81	47.48	83.75	−0.649	Nucleus
Sne01G037860.1	280	31,202.13	9.72	55.83	62.39	−0.739	Nucleus
Sne05G021620.1	304	33,834.93	6.08	55.27	64.21	−0.752	Nucleus
Sne05G068940.1	285	32,060.75	5.54	60.29	69.47	−0.773	Nucleus
Sne05G062180.1	280	31,751.71	4.84	64.12	75.21	−0.627	Nucleus
Sne05G046630.1	788	88,406.95	4.83	57.52	68.07	−1.084	Nucleus
Sne04G020830.1	297	33,785.70	6.67	56.11	63.10	−0.753	Nucleus
Sne05G010480.1	257	28,740.63	6.34	68.82	49.38	−0.981	Nucleus
Sne01G048940.1	192	22,453.08	6.13	62.03	63.07	−0.931	Nucleus
Sne01G039770.1	407	44,374.73	5.89	45.03	69.04	−0.509	Nucleus
Sne05G015190.1	236	26,827.55	6.24	54.32	71.61	−0.598	Nucleus
Sne05G065800.1	190	21,454.17	5.52	55.76	77.63	−0.725	Nucleus
Sne05G067030.1	260	29,540.09	5.26	55.16	69.04	−0.790	Nucleus
Sne03G031190.1	305	34,465.41	6.07	58.64	72.30	−0.710	Nucleus
Sne05G081950.1	320	35,714.81	6.24	61.93	66.25	−0.724	Nucleus
Sne06G027900.1	486	53,644.72	5.63	52.96	62.04	−0.654	Nucleus
Sne05G053870.1	219	25,053.46	7.64	62.88	74.38	−0.723	Nucleus
Sne04G031160.1	267	29,945.79	6.80	54.33	70.97	−0.686	Nucleus
Sne03G027380.1	238	27,716.88	8.90	34.29	61.05	−0.886	Nucleus
Sne06G026210.1	527	57,194.51	5.48	53.10	65.37	−0.547	Nucleus
Sne01G052060.1	369	39,510.17	5.04	44.42	66.94	−0.485	Nucleus
Sne03G013390.1	312	33,831.93	6.15	40.15	73.78	−0.605	Nucleus
Sne05G081350.1	348	38,947.80	6.83	51.22	68.16	−0.705	Nucleus
Sne01G018250.1	350	40,411.70	6.09	44.14	59.97	−0.952	Nucleus
Sne05G056850.1	278	29,973.27	5.11	67.47	61.55	−0.517	Nucleus
Sne03G022110.1	189	21,711.74	10.05	69.80	63.07	−0.687	Nucleus
Sne04G029970.1	204	23,701.70	9.03	56.52	68.38	−0.780	Nucleus
Sne06G020380.1	265	30,853.05	6.26	50.12	55.32	−0.961	Nucleus
Sne01G024350.1	286	31,741.88	8.64	68.94	64.90	−0.450	Nucleus
Sne06G025470.1	547	59,646.42	5.93	52.60	67.42	−0.604	Nucleus
Sne02G001930.1	254	29,056.43	5.29	67.78	66.02	−0.867	Nucleus
Sne05G008390.1	362	38,983.17	6.39	52.80	64.31	−0.638	Nucleus
Sne06G023080.1	238	27,171.75	9.33	73.66	53.28	−0.830	Nucleus
Sne03G020330.1	279	31,405.14	6.55	52.93	75.56	−0.512	Nucleus
Sne04G009720.1	196	21,707.75	4.46	62.54	58.27	−0.615	Nucleus
Sne05G053750.1	207	24,226.84	9.22	64.94	87.05	−0.778	Nucleus
Sne06G004680.1	232	27,095.97	5.50	65.09	54.27	−0.976	Nucleus
Sne05G037810.1	288	32,349.53	8.83	46.04	72.88	−0.694	Nucleus
Sne04G034820.1	229	25,644.65	5.32	63.62	67.82	−0.649	Nucleus
Sne01G002340.1	267	30,215.82	6.72	55.86	64.72	−0.755	Nucleus
Sne05G068720.1	324	36,000.75	9.20	53.52	76.20	−0.577	Nucleus
Sne05G043040.1	275	31,851.61	9.62	50.81	59.93	−0.973	Nucleus
Sne06G004080.1	246	27,752.56	9.75	53.61	53.50	−0.972	Nucleus
Sne02G011360.1	232	26,203.08	10.21	40.67	67.28	−0.733	Nucleus
Sne05G039570.1	241	28,043.30	7.05	65.55	65.64	−0.925	Nucleus
Sne03G022180.1	278	31,708.10	8.38	48.63	73.74	−0.677	Nucleus
Sne05G053730.1	263	29,122.95	8.77	72.65	68.37	−0.627	Nucleus
Sne02G005200.1	281	32,102.15	6.10	67.05	61.14	−0.658	Nucleus
Sne05G040480.1	241	28,134.38	6.41	65.90	64.02	−0.949	Nucleus
Sne01G017620.1	283	31,447.23	6.24	50.78	72.12	−0.608	Nucleus
Sne05G076150.1	271	30,762.60	6.14	58.58	64.80	−0.866	Nucleus
Sne05G036630.1	217	24,355.92	9.02	47.97	74.70	−0.653	Nucleus
Sne06G017650.1	367	42,412.80	5.85	68.14	54.25	−1.005	Nucleus
Sne05G024190.1	339	38,587.92	9.45	71.35	70.65	−0.898	Nucleus
Sne06G014590.1	207	23,365.44	8.79	49.10	64.59	−0.801	Nucleus
Sne05G085850.1	243	26,162.26	9.35	61.86	63.83	−0.544	Nucleus
Sne03G025760.1	262	28,967.25	9.43	62.76	79.47	−0.569	Nucleus
Sne05G021820.1	367	41,827.11	8.82	66.22	74.96	−0.774	Nucleus
Sne01G017290.1	260	29,530.35	8.11	48.25	76.19	−0.667	Nucleus
Sne05G024220.1	518	57,840.01	8.60	53.94	70.60	−0.717	Nucleus
Sne05G009240.1	303	33,979.27	5.35	62.50	63.99	−0.598	Nucleus
Sne01G048800.1	292	32,345.11	8.60	42.96	65.86	−0.634	Nucleus
Sne04G014800.1	991	110,231.84	5.05	58.95	59.55	−0.792	Nucleus
Sne06G017930.1	224	25,609.83	6.67	55.57	68.79	−0.682	Nucleus
Sne03G031200.1	317	35,418.66	5.82	46.00	78.52	−0.594	Nucleus
Sne04G027630.1	328	36,868.69	5.21	40.21	79.09	−0.518	Nucleus
Sne05G088960.1	275	30,674.52	6.67	56.98	67.85	−0.663	Nucleus
Sne03G031270.1	247	28,395.56	5.78	43.99	62.43	−0.713	Nucleus
Sne06G013090.1	202	22,682.28	9.24	52.83	65.25	−0.786	Nucleus
Sne05G053810.1	219	25,088.38	6.22	53.62	69.91	−0.771	Nucleus
Sne01G019030.1	298	32,877.90	6.59	46.25	66.21	−0.647	Nucleus
Sne05G087990.1	332	36,951.56	8.38	56.63	72.38	−0.575	Nucleus
Sne05G009800.1	193	22,104.76	6.96	58.46	64.72	−0.869	Nucleus
Sne05G033540.1	273	29,792.27	6.46	54.79	74.10	−0.603	Nucleus
Sne05G053740.1	314	35,248.92	8.92	72.30	63.76	−0.703	Nucleus
Sne04G047920.1	331	37,432.90	5.89	52.31	81.03	−0.663	Nucleus
Sne05G069840.1	318	34,559.64	9.65	77.72	64.09	−0.657	Nucleus
Sne03G030810.1	243	27,784.20	6.54	66.54	66.30	−0.759	Nucleus
Sne03G027030.1	300	32,807.28	6.71	56.05	76.77	−0.472	Nucleus
Sne03G012760.1	229	25,337.30	6.12	48.63	73.28	−0.645	Nucleus
Sne05G077440.1	232	26,379.49	5.40	54.64	67.24	−0.683	Nucleus
Sne05G025370.1	327	36,610.32	6.85	57.20	72.57	−0.458	Nucleus
Sne05G008470.1	299	33,662.26	6.31	54.21	52.68	−0.902	Nucleus
Sne05G066770.1	228	25,325.94	9.72	60.82	76.75	−0.591	Nucleus
Sne04G023680.1	250	27,757.21	6.44	56.24	74.24	−0.450	Nucleus
Sne05G081170.1	305	34,466.52	7.73	54.64	61.84	−0.775	Nucleus

## Data Availability

Data are contained within the article.

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
