# Peer review of "Genome-Wide Identification and Expression Analysis of the MYB Transcription Factor Family in Salvia nemorosa"

_genes, 2024, doi:10.3390/genes15010110_

Round 1

Reviewer 1 Report

Comments and Suggestions for Authors

The manuscript is prepared on an interesting topic, but it contains fundamental shortcomings that need to be corrected or supplemented with fundamental methodological information. Keywords - be careful there is a big match with the title of the manuscript. Introduction - the authors mention the significant interaction of MYB and bHLH transcription factors. I recommend to add the information that even bHLHs play an important role in the coloring of flowers or fruits (see Muhammad et al. 2023 - DOI: 10.17221/2/2022-CJGPB). The material and methods are very elaborately processed. authors use formulations - our DNA and cDNA databases, etc. It is not possible to refer to source data that are not accessible to readers. In methods, it must be written clearly and reproducibly without unknowns that can significantly distort the results achieved (otherwise it is not possible to trust the results achieved even in the context of bioinformatics analyses). When we get to the methodology of our own experiment, unspecified material appears here (P1-P4). It is necessary to supplement and perform its basic characteristics! Similarly, specify the selection of candidate genes. The choice must be logically justified, so that the achieved results are also understandable. The link to some (unspecified) study is misleading! The manuscript contains a link to Supplements that are not part of the manuscript (?incomplete submission?). Due to these shortcomings, it is not possible to adequately comment on the achieved results and their discussion. This is only possible after completing and eliminating methodological shortcomings.  I believe that if the authors carefully rework, especially the methodological part, the manuscript can bring interesting and up-to-date knowledge.

Author Response

Q1. The manuscript is prepared on an interesting topic, but it contains fundamental shortcomings that need to be corrected or supplemented with fundamental methodological information. Keywords - be careful there is a big match with the title of the manuscript. Introduction - the authors mention the significant interaction of MYB and bHLH transcription factors. I recommend to add the information that even bHLHs play an important role in the coloring of flowers or fruits (see Muhammad et al. 2023 - DOI: 10.17221/2/2022-CJGPB).

Response: Thank you for your improvement and recommendations. Specifically, we have incorporated the reports on the involvement of the MYB-bHLH-WD40 complex in regulating apocynthion biosynthesis, with particular emphasis on the role of the transcription factor bHLH as outlined on line 85-89, page 2.

Q2. The material and methods are very elaborately processed. authors use formulations - our DNA and cDNA databases, etc. It is not possible to refer to source data that are not accessible to readers. In methods, it must be written clearly and reproducibly without unknowns that can significantly distort the results achieved (otherwise it is not possible to trust the results achieved even in the context of bioinformatics analyses).

Response: Thank you for the suggestions from the reviewers. The genome of Salvia nemorosa used in this study was sequenced, assembled, and annotated in our laboratory. The sequencing data of the S. nemorosa genome has been deposited in a public database, and the paper is presently undergoing the submission process. In the near future, the data will be made available for download, allowing more researchers to access the sequencing data.

Q3. When we get to the methodology of our own experiment, unspecified material appears here (P1-P4). It is necessary to supplement and perform its basic characteristics!

Response: Thank you for the suggestions from the reviewer. Based on your advice, we have added a description of the samples in the methods section, as seen at the Section 2.5 on line 189-191, page 4.

Q4. Similarly, specify the selection of candidate genes. The choice must be logically justified, so that the achieved results are also understandable.

Response: We apologize for the lack of clarity in the methods section. This article identifies the mouse tail grass MYB family at the genome level using popular gene family identification methods including HMM, Blast, and manual identification of CDD domains. The sequences were then subjected to in-depth analysis. We have rewritten the Method Section 2.1 in line 133-138, page 3.

Q5. The link to some (unspecified) study is misleading! The manuscript contains a link to Supplements that are not part of the manuscript (?incomplete submission?). Due to these shortcomings, it is not possible to adequately comment on the achieved results and their discussion. This is only possible after completing and eliminating methodological shortcomings.  I believe that if the authors carefully rework, especially the methodological part, the manuscript can bring interesting and up-to-date knowledge.

Response: We regret to inform you that the supplement file is currently unavailable for display due to unforeseen circumstances. We will promptly re-upload the supplement file, which will include primer information and ###. Thank you for your understanding.

Reviewer 2 Report

Comments and Suggestions for Authors

Dear Authors

Review of the Paper titled: "Analysis of MYB Genes in Salvia nemorosa in the Context of Anthocyanin Biosynthesis"

Detailed Remarks:

1. Abstract is Well-Structured.

2. The abstract is well-structured, providing a concise overview of the research.

3. Introduction Provides Sufficient Context:

4. The introduction adequately introduces the topic and concludes with a clear research objective.

5. A recommendation is made for the authors to consider stating an alternative hypothesis in addition to the null hypothesis, enhancing the research design.

6. The materials and methods section is sufficiently presented.

7. Results and Discussion Need Improvement,

Strengths:

1. Comprehensive Analysis: The authors conducted a comprehensive analysis of MYB genes in Salvia Nemorosa, covering identification, phylogeny, gene structure, chromosomal localization, cis-acting elements, and expression profiles. This multifaceted approach provides a thorough understanding of various aspects of these genes in the context of anthocyanin biosynthesis.

2. Use of Various Techniques: The text employs various research techniques, such as phylogenetic analysis, three-dimensional protein structure prediction, motif and gene structure analysis, chromosomal localization, and prediction of cis-acting elements. This multi-faceted approach allows for a fuller picture.

3. Demonstration of Potential Role in Anthocyanin Biosynthesis: Identification of MYB genes clustered with those known for anthocyanin regulation suggests their potential role in anthocyanin biosynthesis in Salvia nemorosa. This serves as a significant starting point for further functional studies.

Weaknesses:

1. Lack of Experimental Confirmation: Despite identifying potentially key genes, the text does not provide experimental confirmations of their role in anthocyanin biosynthesis. Therefore, direct functional evidence from in vivo or in vitro experiments is missing.

2. Absence of Mention of Potential Limitations: The authors do not mention potential limitations of their study, such as the scope of experimental data, which could impact the overall reliability of results.

3. Complex Technical Language: The text is highly technical and requires a certain level of biological and bioinformatic knowledge, making it potentially challenging for readers outside the biological sciences.

Summary: The paper offers an in-depth analysis of MYB genes in the context of anthocyanin biosynthesis in Salvia nemorosa. However, the lack of experimental confirmations and the absence of consideration for potential study limitations pose challenges. The complex language may also hinder understanding for readers outside the field. It is recommended to provide experimental evidence for the functionality of identified genes and address potential study limitations.

Potential Areas for Improvement:

1. Lack of Experimental Confirmations: In the discussion, there could be a stronger emphasis on the need for experimental confirmations of the functions of identified genes. Indicating the necessity for future in vivo or in vitro experiments would be crucial.

2. Providing More Biological Context: In some parts of the discussion, especially in the interpretation of gene expression, providing more biological context could be beneficial. For instance, explaining why certain genes exhibit higher expression levels at specific flowering stages.

3. Consideration of Potential Limitations: Discussing potential limitations of the study should be included in the discussion. Mentioning potential limitations related to data, analytical techniques, or study scope would contribute to a more comprehensive discussion.

4. Explanation of Result Discrepancies: In cases where results from gene expression analysis seem contradictory (e.g., heatmap vs. qRT-PCR), providing a more detailed explanation of these differences would be valuable.

In conclusion, while the interpretation of results and the discussion contain many positive aspects, there are several areas for improvement, especially in emphasizing the need for experimental confirmations and providing additional biological context.

Comments on the Quality of English Language

Minor editing of English language required

Author Response

Thank you for your comments and suggestions on this article. Based on your suggestions, we have provided a response to “Potential Areas for Improvement” and made revisions to the entire article as per your advice.

  1. Lack of Experimental Confirmations: In the discussion, there could be a stronger emphasis on the need for experimental confirmations of the functions of identified genes. Indicating the necessity for future in vivo or in vitro experiments would be crucial.

Response: We appreciate the reviewer for highlighting a limitation of this article. The study utilized bioinformatics methods to identify and analyze the characteristics of 142 MYB genes in S. nemorosa, including their gene structure, chromosomal location, physicochemical properties, expression patterns, and promoter elements. This data serves as a valuable resource for future research on MYB genes in S. nemorosa. Indeed, it is acknowledged that the article did not involve additional experimental validation, and these limitations were addressed in the discussion (seen as line 404-412, page 16). Our forthcoming efforts will involve the selection of genes for stable overexpression, gene editing, and silencing based on the analysis and prediction presented in this study. Additionally, we plan to utilize transient expression to investigate the transcriptional regulatory activity of MYB transcription factors on downstream anthocyanin biosynthetic enzyme genes. These tasks are currently underway.

  1. Providing More Biological Context: In some parts of the discussion, especially in the interpretation of gene expression, providing more biological context could be beneficial. For instance, explaining why certain genes exhibit higher expression levels at specific flowering stages.

Response: Thank you for the reviewer's comments. In response to the suggestions, we have thoroughly examined the roles of the MYB gene across various species, such as apple, petunia, Camellia nitidissima, peony, tulips within the discussion section, aiming to offer a comprehensive biological context.

  1. Consideration of Potential Limitations: Discussing potential limitations of the study should be included in the discussion. Mentioning potential limitations related to data, analytical techniques, or study scope would contribute to a more comprehensive discussion.

Response: We appreciate your feedback regarding the limitations of this article. It is acknowledged that as a literature review, the article lacks coverage of experimental validation. Nevertheless, our ongoing research involves conducting functional validation experiments on the identified candidate genes, encompassing stable transformation and transcriptional regulation mechanisms. This endeavor aims to furnish additional experimental data for the investigation of flower color formation in snapdragons. We have duly addressed this matter in the discussion section.

  1. Explanation of Result Discrepancies: In cases where results from gene expression analysis seem contradictory (e.g., heatmap vs. qRT-PCR), providing a more detailed explanation of these differences would be valuable.

Response: We appreciate your identification of the concerns. Owing to technical limitations, certain genes may exhibit erroneous overexpression or underexpression during transcriptome sequencing due to amplification bias, necessitating additional qPCR validation. Discrepancies between transcriptome sequencing and qPCR outcomes may also stem from sample-related factors. Consequently, we conducted a repeat of the qPCR experiment, yielding results consistent with the manuscript, thus affirming the greater reliability of the PCR findings in our investigation.

Round 2

Reviewer 1 Report

Comments and Suggestions for Authors

The authors accepted most of my comments. In terms of content, it is necessary to supplement the notified supplements TableS1 and FileS1-S3. Without this and the possibility to preview the given files, it is not possible to accept the manuscript, i.e. therefore, I recommend publishing after major revision and second review (checking the contents of the supplements). On the formal side, I am not sure about the designation of the subsection on line 205 + it is not ideal for the subsection to begin with Figure, to which there is a link in the text right after it. I recommend moving this image below i.e. for reference in the text.

After a quick review of the content of the alternates, nothing should prevent the manuscript from being accepted and published.

Author Response

Dear reviewer,

We are grateful for your valuable suggestions. In response, we have repositioned Figure 1 to the conclusion of the manuscript and included Table S1, S2 and the supplementary files. We sincerely acknowledge the reviewer's input in enhancing the quality of this article.

Reviewer 2 Report

Comments and Suggestions for Authors

Dear Authors,

The authors provided comprehensive explanations and made improvements to the paper based on the reviewer's suggestions, including:

1. Utilized bioinformatic methods to identify and analyze the characteristics of 142 MYB genes in S. nemorosa, including their gene structure, chromosomal location, physicochemical properties, expression patterns, and promoter elements. This constitutes a valuable resource for future research on MYB genes in S. nemorosa. Limitations related to this aspect of the work were addressed in the discussion.

2. Provided greater biological context by thoroughly examining and discussing the role of the MYB gene in various species in the discussion section, aiming to present a comprehensive biological context.

3. Considered potential limitations of the study: The authors discussed the implementation of functional verification experiments on identified candidate genes, including stable transformation mechanisms and transcriptional regulation, in the discussion section.

4. Explained discrepancies: The authors demonstrated that due to technical limitations, some genes may exhibit erroneous overexpression or insufficient expression during transcriptome sequencing due to amplification errors. Consequently, they repeated the qPCR experiment, confirming the increased reliability of PCR results in their research.

Comments on the Quality of English Language

Minor English editing required.

Author Response

Dear reviewer,

We sincerely appreciate your suggestions and for helping to improve our manuscript.

Best regards,

Huan Yang